# Exploring whole-body kinematics when eating real foods with the dominant hand in healthy adults

Jun Nakatake[1]*, Koji Totoribe[2], Hideki Arakawa[1,3], Etsuo Chosa[1,3]

**1** Rehabilitation Unit, University of Miyazaki Hospital, Miyazaki, Japan, **2** Department of Rehabilitation, Miyazaki City Tano Hospital, Miyazaki, Japan, **3** Department of Orthopaedic Surgery, University of Miyazaki, Miyazaki, Japan

* jyun_nakatake@med.miyazaki-u.ac.jp

**Data Availability Statement:** All relevant data are within the paper and its Supporting Information files.

**Funding:** This study was partly funded by a grant from the University of Miyazaki Hospital. The

## Abstract

Despite the importance of eating movements to the rehabilitation of neurological patients, information regarding the normal kinematics of eating in a realistic setting is limited. We aimed to quantify whole-body three-dimensional kinematics among healthy individuals by assessing movement patterns in defined phases while eating real food with the dominant hand in a seated position. Our cross-sectional study included 45 healthy, right-hand dominant individuals with a mean age of 27.3 ± 5.1 years. Whole-body kinematics (joint angles of the upper limb, hip, neck, and trunk) were captured using an inertial sensor motion system. The eating motion was divided into four phases for analysis: reaching, spooning, transport, and mouth. The mean joint angles were compared among the phases with Friedman's analysis of variance. The maximum angles through all eating phases were 129.0° of elbow flexion, 32.4° of wrist extension, 50.4° of hip flexion, 6.8° of hip abduction, and 0.2° of hip rotation. The mean shoulder, elbow, and hip joint flexion angles were largest in the mouth phase, with the smallest being the neck flexion angle. By contrast, in the spooning phase, the shoulder, elbow, and hip flexion were the smallest, with the largest being the neck flexion angle. These angles were significantly different between the mouth and spooning phases (p < 0.008, Bonferroni post hoc correction). Our results revealed that characteristic whole-body movements correspond to each phase of realistic eating in healthy individuals. This study provides useful kinematic data regarding normal eating movements, which may inform whole-body positioning and movement, improve the assessment of eating abilities in clinical settings, and provide a basis for future studies.

## Introduction

Rehabilitation of eating is an important component of the care of patients with neurological impairments, such as individuals with cerebral palsy or who have had a stroke [1, 2]. Eating-related impairments, which are not limited to dysphagia, have a negative impact on physical and psychosocial well-being and participation within the wider context of culture and the environment. Various interventional approaches have been used to address eating impairments,

funder had a role in the preparation of the manuscript. There was no additional external funding received for this study.

**Competing interests:** The authors have declared that no competing interests exist.

including environmental changes, adequate positioning, use of adaptive equipment, rehabilitation of motor components of chewing and swallowing, and education of patients and caregivers [3, 4].

Eating is usually performed in a seated position and requires movement of the dominant upper limb, neck, and trunk, with the lower limbs and trunk providing stability. Knowledge of whole-body kinematics, including posture, minimum-maximum joint angles, and movement patterns, would be important to assess impairments and to inform intervention. Furthermore, capturing this information in near-natural settings would be useful, as the kinematics of eating movements in daily living are different from those measured in simulated tasks [5].

The upper limb joint angles in an unimpaired population when eating with a spoon have previously been reported using three-dimensional motion analysis [6–13]. The neck, lumbar spine, and trunk angles have also been studied, although to a lesser extent [6, 9, 14, 15]. Beaudette and Chester [6] and Klotz et al. [9] comprehensively described the upper limb, trunk, and neck angles during realistic eating. However, eating-related whole-body kinematics that include the lower limb, whose position as a stable base of upper limb motion should be assessed in practice, have not been reported in detail.

Evaluation of the change in joint angles over time is important to identify kinematic patterns of eating. Waveform analyses of upper limb joint movements, which provide only a subjective visual assessment of upper limb motion for eating, have been previously reported [6, 11, 13]. However, the time-dependent kinematics of the neck, trunk, and lower limbs have not been clearly defined. Moreover, to the best of our knowledge, no comparison of single points or phases during eating with respect to objective parameters of whole-body kinematics has been reported. Realistic eating consists of specific phases that require concurrent whole-body movements. Due to the complexity of these realistic movements, clinicians in practice may have a limited understanding of the normal kinematics of eating. Establishing whole-body kinematic variables and movement patterns, which could be described as differences in joint positions, the extent of motion, and motion directions between eating phases in patients with eating limitations, would therefore constitute useful information that could be applied in clinical settings. Thus, our study aimed to determine the whole-body joint angles and movement patterns necessary for realistic eating in healthy individuals.

## Materials and methods

### Participants

The inclusion criteria for this cross-sectional study were as follows: 1) age 20–39 years; 2) right-handed; and 3) no neurological, sensory, or musculoskeletal abnormalities. Individuals who used the left hand to eat with a spoon were excluded. Fifty participants (23 men, 27 women; age, mean 27.3, standard deviation [SD] 5.0 years; height, mean 164.7, SD 8.2 cm) were recruited between April 2013 and October 2017 from among the medical staff of our institution, who responded of their own free will to a request for study participants. To ensure normal kinematic data and exclude age effects resulting from neurological immaturity or decline, children [6, 16] and older adults [17] were not selected as their upper limb reaching movements during activities of daily living differ from those of young adults. Left-handers were excluded since manipulations of their dominant hand may not simply mirror those of right-handers [18].

### Instruments and measurements

Three-dimensional joint angle data were acquired in the rehabilitation room of our institution using the Xsens MVN system (Xsens Technologies B.V., The Netherlands), as previously

described [19], and captured at a frequency of 120 Hz. This system contains 17 inertial sensor units (ISUs) and two Xbus Masters (Xsens Technologies B.V.). The kinematic parameters of the eating movement were calculated from raw ISU data using built-in software (Xsens MVN Studio 3.1, Xsens Technologies B.V.). The whole-body model was defined by 23 kinematic data points, with axes and origins defined using the recommendations of the International Society of Biomechanics [20, 21]. Of these points, one neck, four spine, and two toe points were not directly measured but were calculated using ISU data and other data points with respect to global and local references. Joint angles were defined by the position of the distal segment relative to the proximal segment. The neutral (zero) position in joint angles for the model was defined as the joint angles when standing upright, in a relaxed posture, with feet parallel, one-foot width apart, upper limbs alongside the body, palms facing forward, and the head oriented forward. The accuracy of the Xsens MTx and MVN systems used has previously been confirmed against an optical system [22–26].

Eating yogurt with a spoon was chosen as it represents a widely used eating behavior, namely moving the spoon from the bowl to the mouth for eating through movement of the upper limb. One spoon (length, 17.5 cm; weight, 41 g; stainless steel) and bowl (top diameter, 15.5 cm; bottom diameter, 7.0 cm; height, 4.5 cm; weight, 246 g; ceramic) were used.

The participants donned a Lycra suit with attached ISU sensors. After calibration and definition of body dimensions, participants sat comfortably without trunk fixation to allow free motion on a 40-cm high seat behind a table (10 cm from the participant's trunk, at the height of the right elbow), with the right upper limb placed alongside the body at baseline and their feet fully on the floor. To standardize the relative position between the participants and the bowl, a participant's right shoulder, forearm, wrist, and finger joints were placed in the anatomically neutral position, with the right elbow in 90˚ flexion and the right hand placed on the table. The center of the bowl was aligned with the midline of the body for each participant. Participants held a spoon with their right hand and received the task instruction, "please eat three spoons full of yogurt without a break, using habitual whole-body movements and speed, not looking away, and with your left hand resting on your left thigh." For realistic conditions, the eating movements were performed with no absolute start and end positions defined.

## Data processing

One eating cycle was defined as the movement of the spoon from the bowl to the mouth and was subdivided into four phases by visually partitioning the recorded movie: reaching (moving the spoon to the bowl), spooning (getting yogurt into the spoon), transport (moving the spoon from the bowl to the mouth), and mouth (placing the yogurt into the mouth). These phases were subsequently filtered to identify phase recordings that were suitable for further analyses. This cycle was chosen as a realistic eating movement according to the study aims but was not completely realistic because using the non-dominant hand to hold the bowl or to gather food was not permitted. Up to three sequential movement cycles were recorded to provide participants with the opportunity to produce two cycles with all four eating phases. The first cycle, including all four standard phases, was included in the between-subject analysis. Cycles containing the following motions were not accepted because they conflict with our definition of an eating movement: excessive elevation of the upper limb during transport in an exaggerated or unnecessary manner, looking away from the bowl, extraneous movement of the head along the sagittal plane during reaching, more than one spoon movement in the bowl, using the spoon to separate the yogurt before spooning, and shaking yogurt off the spoon during transport. The data of participants unable to produce all four standard phases were also excluded.

**Table 1. Whole-body joint angles across the four eating phases.**

| Joint | Motion direction (+)/(-) | Maximum angle | | Minimum angle | |
|---|---|---|---|---|---|
| | | Mean (SD) | 95% CI | Mean (SD) | 95% CI |
| **Shoulder** | Flexion/Extension | 42.0 (11.6) | 38.5–45.5 | 17.0 (8.4) | 14.5–19.5 |
| | Abduction/Adduction | 41.0 (9.7) | 38.0–43.9 | 26.2 (7.4) | 24.0–28.4 |
| | Internal/External rotation | 13.0 (9.6) | 10.1–15.9 | -0.5 (9.0) | -3.2–2.2 |
| **Elbow** | Flexion/Extension | 129.0 (9.4) | 126.2–131.8 | 95.7 (10.2) | 92.6–98.7 |
| **Forearm** | Pronation/Supination | 103.7 (18.8) | 98.0–109.3 | 22.9 (15.9) | 18.2–27.7 |
| **Wrist** | Flexion/Extension | -16.7 (15.2) | -21.3–-12.1 | -32.4 (12.7) | -36.2–-28.6 |
| | Radial/Ulnar deviation | 4.6 (10.0) | 1.6–7.6 | -13.9 (7.5) | -16.1–-11.6 |
| **C7–T1** | Flexion/Extension | 26.2 (3.6) | 25.1–27.3 | 16.9 (5.7) | 15.2–18.6 |
| | Right/Left lateral flexion | 4.8 (3.3) | 3.8–5.8 | -1.7 (2.0) | -2.3–-1.1 |
| | Left/Right rotation | 0.4 (1.6) | -0.1–0.9 | -4.0 (2.8) | -4.8–-3.2 |
| **T8–T9** | Flexion/Extension | 11.9 (4.1) | 10.7–13.1 | 10.5 (3.9) | 9.3–11.6 |
| | Right/Left lateral flexion | 0.3 (0.8) | 0.0–0.5 | -0.4 (0.8) | -0.6–-0.1 |
| | Left/Right rotation | -0.6 (0.7) | -0.8–-0.4 | -1.2 (0.9) | -1.4–-0.9 |
| **L5–S1** | Flexion/Extension | 14.5 (4.2) | 13.2–15.8 | 12.7 (4.2) | 11.4–13.9 |
| | Right/Left lateral flexion | 0.6 (1.0) | 0.3–1.0 | -0.2 (1.1) | -0.5–0.1 |
| | Left/Right rotation | -0.6 (0.8) | -0.9–-0.4 | -1.4 (1.0) | -1.7–-1.1 |
| **Hip** | Flexion/Extension | 50.4 (10.9) | 47.1–53.7 | 44.1 (10.1) | 41.1–47.1 |
| | Abduction/Adduction | 6.8 (10.9) | 3.5–10.0 | 5.7 (10.9) | 2.4–9.0 |
| | Internal/External rotation | 0.2 (6.3) | -1.6–2.1 | -1.2 (6.0) | -3.0–0.6 |

All values are reported in degrees. CI, confidence interval; SD, standard deviation.

The joint angles and direction of movements used to describe the kinematics of the movement are listed in Table 1. Of these, the C7–T1 joint was selected as the neck joint, whereas the T8–T9 and L5–S1 joints were selected as the spine joints. Joint angles during each phase of the movement were extracted from the kinematic data. Using Microsoft Excel (version 16.0, Microsoft Corporation, Redmond, WA), the between-subject mean, SD, and 95% confidence interval (CI) of the maximum and minimum joint angles were calculated for the four phases of the eating movement. Joint motion directions, in which the difference between the minimum and maximum joint angles was >5.0˚, were identified for further analyses as these are clinically detectable. To compare and characterize the joint angles of each phase, the following variables were calculated: mean angles, calculated from the cumulative sequential data over each phase to provide the average position; range of motion (ROM), calculated as the difference between the minimum and maximum angles in each phase to quantify the motion [27]; and joint angles normalized to 101 points, corresponding to 0–100% of the movement time in each phase. The normalized data were then represented as the mean (SD) for each point in each phase, and waveforms of normalized joint angles were calculated for the visual analysis of the joint motion direction.

## Statistical analysis

The Shapiro–Wilk test was used to determine the normality of minimum and maximum joint angle data distribution with the significance level set at $p < 0.05$. The mean, SD, and 95% CI values were calculated to determine the between-subject consistency in joint angles. Friedman's analysis of variance (ANOVA) ($p < 0.05$) with Bonferroni adjustment for multiple comparisons ($p < 0.008$) was used to compare the mean angles and ROMs among the four phases.

The effect size (r) was calculated and defined as follows: |r| = 0.10–0.29, small effect; |r| = 0.30–0.49, medium effect; and |r| ≥ 0.50, large effect [28]. All statistical analyses were performed using SPSS statistics (version 23, IBM Japan, Japan).

## Results

### Study sample

After screening for exclusion, the data of 45 participants from the 50 enrolled were included: 37 first cycles and 8 second cycles. The relevant characteristics of our study group were as follows: 23 men, mean (SD) age of 27.3 (5.1) years, and mean (SD) height of 164.8 (8.6) cm. Of note, a rest period in upper limb motion, i.e., the reaching movement paused within several seconds and started again, was frequently observed during the reaching phase (26 of 37 first cycles) and was therefore included in the analysis.

### Necessary joint angles for eating movements

Detailed whole-body joint angles for all four phases are summarized in Table 1. A difference of >5.0˚ between the maximum and minimum was identified across all four phases for the shoulder, elbow, forearm, and wrist joint angles, flexion and right lateral flexion angles of C7–T1, and the flexion angle of the right hip joint. Accordingly, the mean angles, ROMs, and waveforms of normalized joint angles in each phase were used for comparison among the phases.

### Comparison of mean angles among eating phases

The mean angles for all four phases are presented in Fig 1. The highest mean angles (upper limb, C7–T1, and hip) were identified in the spooning and mouth phases, except for shoulder internal rotation and wrist flexion. Overall, the mean angles were significantly different across all four phases (Friedman's ANOVA, p < 0.05). Significant between-phase differences after Bonferroni correction with a large effect size (p < 0.008, |r| ≥ 0.5) are described as follows (Fig 1 and S4 Table). At the shoulder, the flexion and abduction mean angles were largest in the mouth phase and larger in the transport than in the reaching and spooning phases, with greater mean internal rotation angles in the spooning, transport, and mouth phases than in the reaching phase. Elbow flexion was largest in the mouth phase and smallest in the spooning phase. The position of forearm pronation decreased from the spooning phase through the reaching, transport, and mouth phases. The mean angle of wrist flexion was greater for the mouth than for the reaching phase, whereas the mean angle of radial deviation was greater for the reaching and spooning phases than for the transport and mouth phases. The largest mean flexion angle at C7–T1 was observed during the spooning phase, with the smallest mean angle during the mouth phase. By contrast, the largest mean angle of right lateral flexion of C7–T1 was detected in the mouth phase, with the mean angle being smaller in the spooning than in the transport and mouth phases. The largest mean hip flexion angle was observed during the mouth phase, with the smallest during the spooning phase.

### Comparison of ROMs among eating phases

The ROMs of the upper extremity, C7–T1, and hip were larger during the reaching and transport phases than during the spooning and mouth phases, except for wrist flexion and radial deviation (Fig 2). Overall, ROMs were significantly different across all four phases (Friedman's ANOVA, p > 0.05). Significant between-phase differences after Bonferroni correction with a large effect size (p < 0.008, |r| ≥ 0.5) are described as follows (Fig 2 and S6 Table). The flexion ROM at the shoulder decreased from the reaching phase to the transport, mouth, and

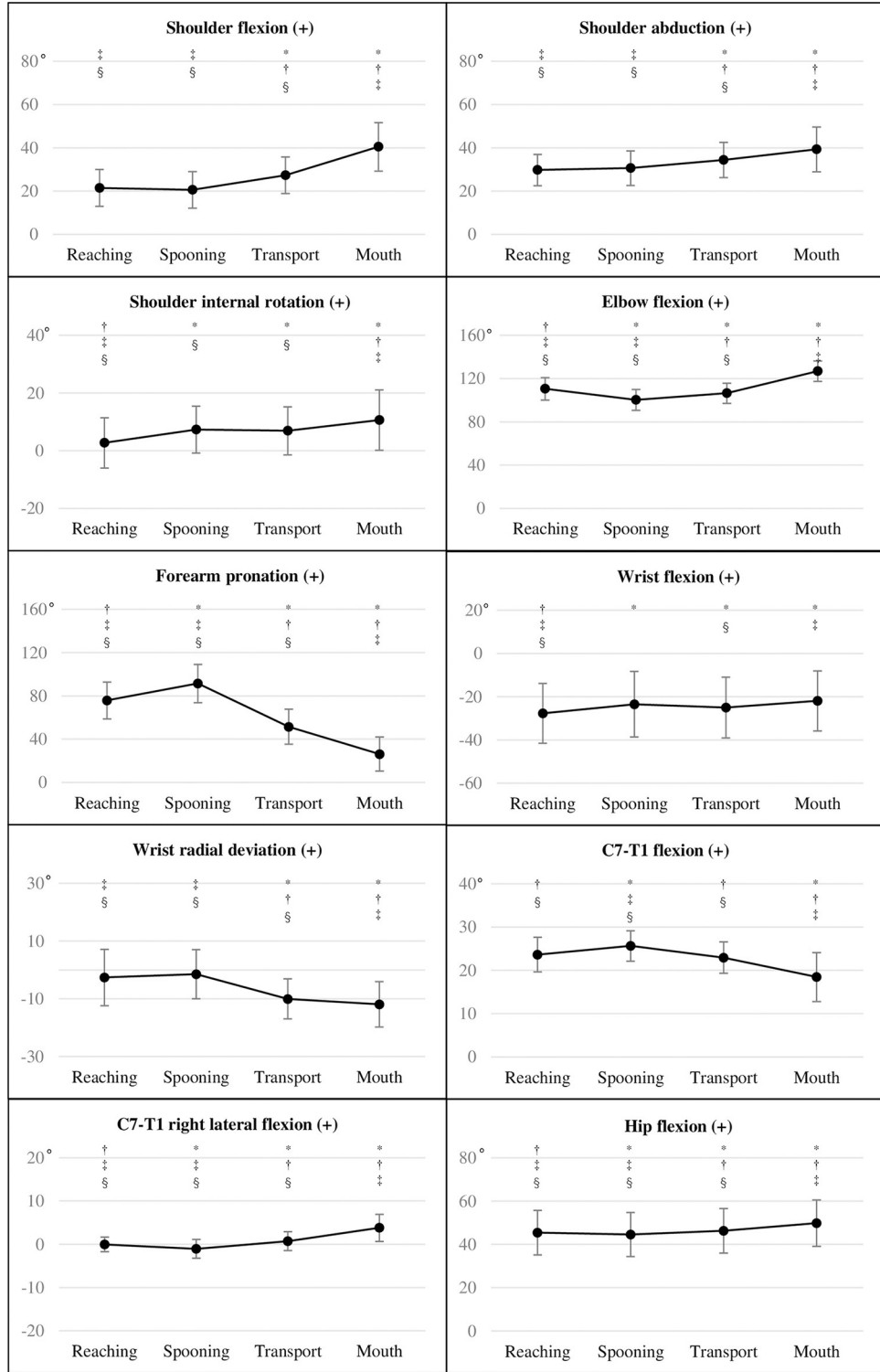

**Fig 1. Mean angles in each phase with Bonferroni correction.** Eating phases are shown on the horizontal axis. *, significant difference (p < 0.008) compared to the reaching phase; †, significant difference (p < 0.008) compared to the spooning phase; ‡, significant difference (p < 0.008) compared to the transport phase; §, significant difference (p < 0.008) compared to the mouth phase.

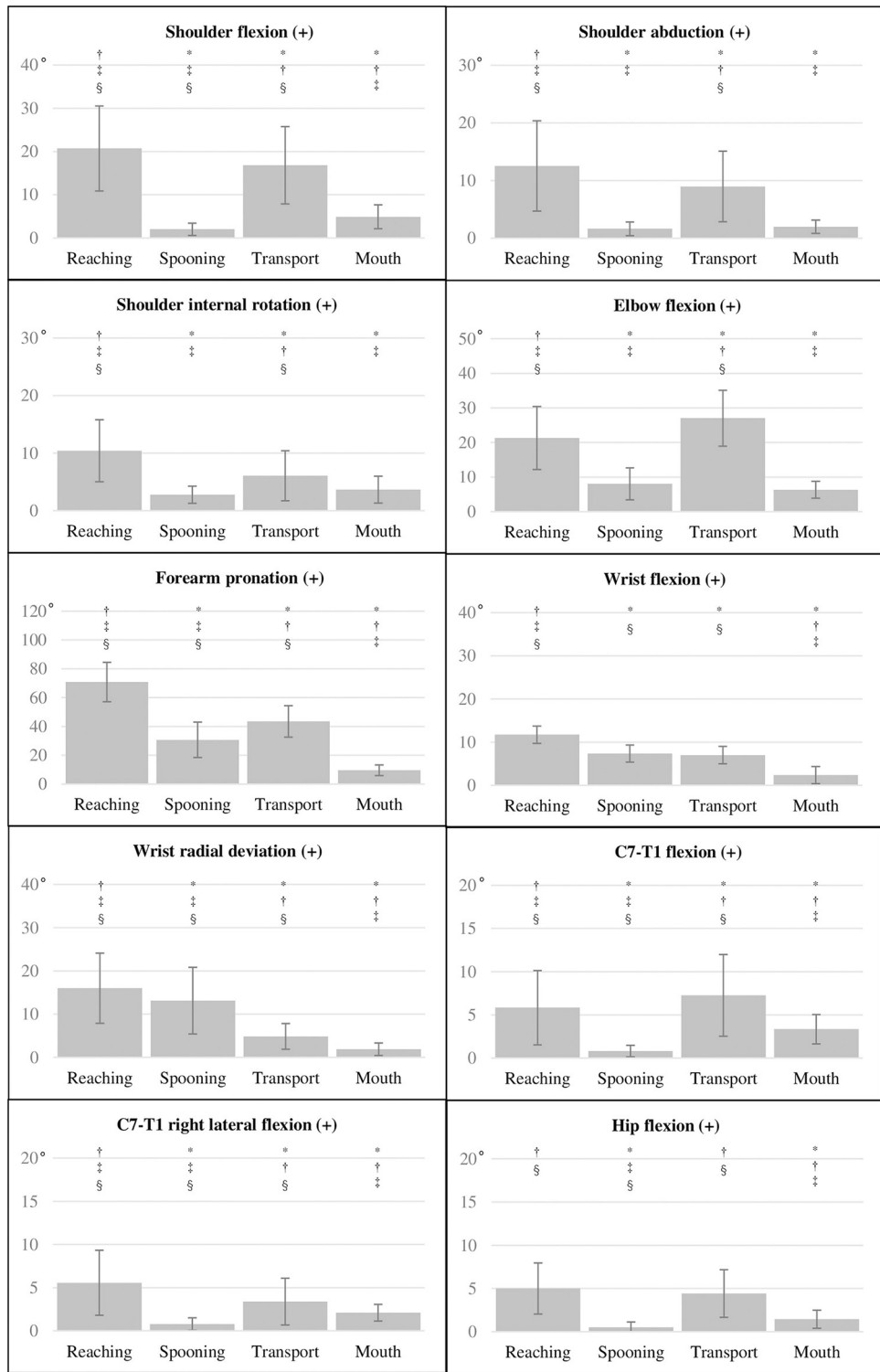

**Fig 2. Ranges of motion in each of the four eating phases with Bonferroni correction.** Eating phases are shown on the horizontal axis. *, significant difference ($p < 0.008$) compared to the reaching phase; †, significant difference ($p < 0.008$) compared to the spooning phase; ‡, significant difference ($p < 0.008$) compared to the transport phase; §, significant difference ($p < 0.008$) compared to the mouth phase.

spooning phases. The ROM of shoulder abduction was greater in the reaching and transport phases than in the spooning and mouth phases, whereas shoulder internal rotation was greatest during the reaching phase. The ROM of elbow flexion was greatest during the reaching and transport phases than in the spooning and mouth phases. The ROM of forearm pronation decreased from the reaching phase through the transport, spooning, and mouth phases, with the difference between the spooning and transport phases being of medium effect size ($p < 0.008$, $r = -0.408$). The ROM of wrist flexion was greater during the reaching, spooning, and transport phases than during the mouth phase, whereas radial deviation was greater during the reaching and spooning phases than during the transport and mouth phases, with the smallest ROM being observed in the mouth phase. The C7–T1 flexion ROM decreased from the transport phase through the reaching, mouth, and spooning phases, with only a medium effect size of the difference between the reaching phase and the transport and mouth phases ($p < 0.008$, $r = -0.367$ and $p < 0.008$, $r = -0.374$, respectively). The C7–T1 right lateral flexion ROM decreased from the reaching phase through the transport, mouth, and spooning phases, with only a small effect size of the difference between the transport and mouth phases ($p < 0.008$, $r = -0.276$). The hip flexion ROM was larger for the reaching and transport phases than for the spooning and mouth phases. Whole-body changes in joint angles over time across all four phases are shown in Fig 3.

## Discussion

We presented an analysis of the three-dimensional whole-body kinematics during a realistic eating movement among healthy young adults and evaluated changes in the movement patterns across the four eating phases. Our findings can provide a reference for clinical assessment and inform the treatment of individuals with neurological impairments affecting the motor control for eating.

A previous study of realistic eating movement reported angles of 130° of elbow flexion, 40° of wrist extension, and 18° of neck flexion [6], with these findings being comparable to ours. In our study, we additionally report the angles of the right hip joint, with a range of 44.1–50.4° of flexion (95% CI, 41.1–53.7°), 5.7–6.8° of abduction (95% CI, 2.4–10.0°), and -1.2–0.2° of rotation (95% CI, -3.0–2.1°). A neutral trunk position has been associated with faster eating movements compared to kyphotic or laterally flexed positions [29]. Other studies have reported greater movement of the trunk and elbow flexion during a drinking task among individuals who have had a stroke compared to that among controls; however, reports of between-group comparisons during eating have been lacking [30, 31]. The trunk position is supported by the lower limbs in seated eating, and the control of the lower limbs is accomplished in the normal hip joint angles. The regulation of the hip angles and the reduction in elbow extension during eating should therefore be investigated in future studies. Although the hip angle varied little among our study group, the compensatory coordination of this position along with reduced elbow extension could have an important clinical effect on eating in populations with neurological problems.

Shoulder flexion, abduction and internal rotation, elbow flexion, and wrist flexion angles have previously been reported to increase when placing food in the mouth and to decrease when moving the hand away from the mouth, with opposite patterns found for forearm pronation and wrist radial deviation [6, 11, 13]. Our findings are consistent with these results, with differences in most mean angles being between the mouth and reaching or transport phases. The large effect sizes of these differences suggest that this is the typical upper limb movement pattern when eating with a spoon.

Beaudette and Chester [6], however, reported a different movement pattern for shoulder joint abduction; this difference likely resulted from their combination of data from both the

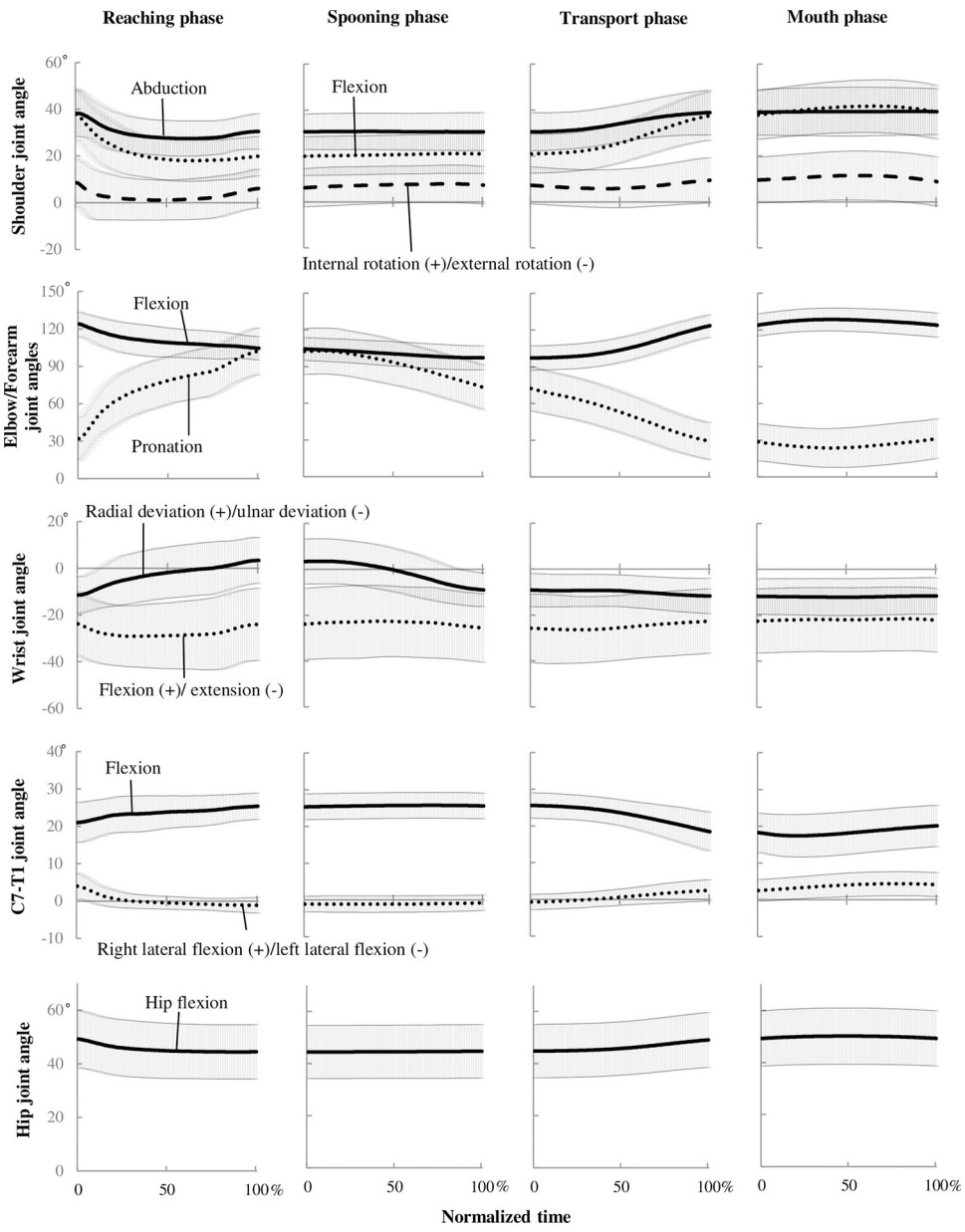

**Fig 3. Changes in whole-body joint angles over time for all four eating phases.** Solid, dotted, and dashed lines show the mean angles, with shaded areas showing the corresponding standard deviations.

dominant and non-dominant hand. In the spooning phase of our study, the mean angles in shoulder flexion and abduction, as well as elbow flexion, were small overall with large mean angles of forearm pronation and wrist radial deviation. Of note, upper limb joint angles in the spooning phases were approximately opposite to those in the mouth phase. The movement patterns of the neck, trunk, and lower limb were specific to each eating phase. Especially, the C7–T1 flexion angle was large, whereas the C7–T1 right lateral flexion and the hip flexion angles were small in the spooning phase, with these angles being respectively reversed in the mouth phase.

We also demonstrated that ROMs of the upper limb, neck, and hip were generally larger during the reaching and transport phases than during the spooning and mouth phases. Combining the results of the mean angles and ROMs, our study highlighted that the upper limb, neck, and hip simultaneously move during the reaching and transport phases, reversing in direction during the spooning and mouth phases. Although the ROMs at these joints are small, the positions of these joints are at their maxima. The different movements from this pattern, especially the reaching phase, and the mean angles of the shoulder internal rotation and the wrist flexion were the smallest. The spooning phase also required large ROMs of forearm pronation (decreasing angles) as well as of wrist flexion (increasing and decreasing angles) and radial deviation (decreasing angles). Knowledge of the movement patterns that we have described may be useful for the comprehensive assessment of eating and optimization of upper and lower limb, neck, and trunk movements during eating. To facilitate or compensate for the normal positions and movements for each eating phase, appropriate positioning, seating equipment, upper extremity and posture control, and orthotics [3, 4] referring to the mean angles, the ROMs, and the changes in joint angles of the present study could be provided in practice.

With regard to the trunk, C7–T1 left rotation, T8–T9 and L5–S1 flexion, lateral flexion, and rotation, as well as hip abduction and rotation were quasi-invariable across the four phases of eating. This may reflect the conditions of our task that used yogurt and required maintenance of the position of the bowl. This is an important finding for practice. Specifically, our standardized task could be used to determine the need for support through equipment or practitioner manipulation in individuals with impairments in the motor control of eating.

The existence of head and trunk motion when placing food in the mouth has been previously reported [32, 33]. Among our study group, the mouth phase was characterized by minimum neck flexion but maximum right neck lateral flexion and hip flexion. These movements are thought to facilitate directing the mouth to the food [4]. The neck and lower limb movement patterns when food approaches the mouth from the left side when eating with the left hand or when food is brought to the mouth using assistance should be addressed in future studies.

Inter-subject differences may have contributed to the high variability in joint angles shown in Fig 3, such that the SD values of all joint angles and phases do not converge. There was no clear difference in movement strategies among the four defined eating phases analyzed in this study, even after the exclusion of outliers with excessive deviation. However, as the aim of the study was to analyze habitual whole-body movements, movements at the participant's own speed without constraints from a backrest were allowed, and the seat height, as well as the bowl and spoon sizes, were not adapted to the participant's body size. The start and stop positions were also defined within uninterrupted eating movements. Thus, measurements of realistic eating behavior can easily vary. The influence of body constitution and muscle strength, which may have caused movement differences in the current study, should be examined in clinical settings by future studies. In contrast, a comparison of eating phases could foster an understanding of typical whole-body movement patterns in a realistic condition, such as that described above, allowing movement patterns that reflect realistic eating to be adopted in clinical practice.

Increased maximal shoulder abduction and reduced maximal elbow flexion of healthy pediatric participants compared to those of healthy young adults were reported previously [6]. Since these maximal angles were shown during the mouth phase in our study with healthy young adults, larger shoulder abduction and lower elbow flexion may be able to be assessed as normal pediatric motions in the mouth phase. Although the effects of aging on eating movements remain unclear, the reaching movement during eating may be affected in older

individuals. Healthy older adults have an increased number of jerky and undirected movements during reaching tasks [17], possibly resulting in increased joint angles of the upper limb compared to our results. Maximal shoulder flexion and abduction angles in individuals with rotator cuff impingement are approximately the same as those in healthy individuals [7]; thus, these two groups' shoulder movements would not be different in the mouth phase examined here. The severe upper extremity function in children with unilateral cerebral palsy and their maximal forearm pronation angle during eating are positively correlated [9], which could be observed in our spooning phase.

The limitations of our study need to be acknowledged. We constrained the task by not allowing the individual to hold the bowl, only used yogurt, and excluded some movements such as the excessive raising of upper limbs during the transporting phase that was not predefined in the study task. Although this standardized our assessment, realistic eating movements still had wide variability as described above. This limitation is important to be considered for any clinical application of our study results. The findings also may not be generalizable to other types of food, utensils other than a spoon, and other eating movements, such as stirring, cutting, and gathering food. In addition, we only investigated young healthy adults who were right-hand dominant. Clinical assessments according to our experimental protocol should be considered, with guidance to use the strict instructions demonstrated in the study. Evaluation of left-handed individuals and other age groups, as well as clinical populations, is warranted considering previous findings of differences in the kinematics of eating due to injury [7, 8], disease [9], individual characteristics [6], and equipment used [11, 12, 33]. The number of participants should also be increased in future studies, and more factors that cause individual eating movement patterns should be investigated. The current study had no preplanned sample size and the study power had not been calculated because the significance level and power of differences in measured joint angle data between phases were not known prior to the study. Finally, validation of our kinematic model and the ISU measurements used would be indicated for reliable implementation in practice.

## Conclusions

Our study regarding the three-dimensional kinematics of eating in healthy young adults highlighted the combined movement of the upper limb, neck, and hip in the same direction during the reaching and transport phases, with a change in the direction of motion at these joints during the spooning and mouth phases. This information can be used in clinical practice to facilitate education on adequate positioning and movements to improve eating, as well as to evaluate the prognosis of patients with eating impairments. Our findings may also provide a base for future kinematic studies on eating in different populations and to inform interventions.

## Supporting information

**S1 Table. Individual values of maximum whole-body joint angles across all four phases of eating.**
(XLSX)

**S2 Table. Individual values of minimum whole-body joint angles across all four phases of eating.**
(XLSX)

**S3 Table. Individual values of mean angles in each phase of eating.**
(XLSX)

**S4 Table. Effect size of differences in mean angles among the four eating phases.**
(XLSX)

**S5 Table. Individual values of ranges of motion in each phase of eating.**
(XLSX)

**S6 Table. Effect size of differences in ranges of motion among the four eating phases.**
(XLSX)

**S7 Table. Time-normalized integrated values of joint angles for each phase of eating.**
(XLSX)

## Acknowledgments

We would like to thank Editage for English language editing.

## Author Contributions

**Conceptualization:** Jun Nakatake, Etsuo Chosa.

**Formal analysis:** Jun Nakatake, Koji Totoribe, Hideki Arakawa.

**Funding acquisition:** Etsuo Chosa.

**Investigation:** Jun Nakatake.

**Methodology:** Jun Nakatake, Hideki Arakawa.

**Project administration:** Etsuo Chosa.

**Resources:** Koji Totoribe.

**Supervision:** Etsuo Chosa.

**Writing – original draft:** Jun Nakatake.

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
