## [Decision Letter · Decision Letter 0]

21 May 2021

PONE-D-20-39538

Exploring whole-body kinematics when eating real foods with the dominant hand in healthy adults

PLOS ONE

Dear Dr. Nakatake,

Thank you for submitting your manuscript to PLOS ONE. After careful consideration, we feel that it has merit but does not fully meet PLOS ONE’s publication criteria as it currently stands. Therefore, we invite you to submit a revised version of the manuscript that addresses the points raised during the review process.

As you can see both reviewers were concerned that you have not articulated clearly the need for this research. If you wish to revise, please carefully address all reviewer concerns, particularly the reason why this work was needed.

We look forward to receiving your revised manuscript.

Kind regards,

Bernadette Ann Murphy, PhD

Academic Editor

PLOS ONE

Journal Requirements:

2. In your Methods section, please provide additional information about the participant recruitment method and the demographic details of your participants. Please ensure you have provided sufficient details to replicate the analyses such as: aa) a description of how participants were recruited, and b) descriptions of where participants were recruited and where the research took place.

3. Please provide a sample size and power calculation in the Methods, or discuss the reasons for not performing one before study initiation.

4. We note that Figure 1 includes an image of a participant in the study. 

5. Thank you for stating in your Funding Statement:

"JN was partly funded by a grant from University of Miyazaki hospital. The funder had a role in preparation of the manuscript."

5.1.  Please provide an amended statement that declares *all* the funding or sources of support (whether external or internal to your organization) received during this study, as detailed online in our guide for authors at http://journals.plos.org/plosone/s/submit-now.  Please also include the statement “There was no additional external funding received for this study.” in your updated Funding Statement.

5.2. Please state what role the funders took in the study.  If the funders had no role, please state: "The funders had no role in study design, data collection and analysis, decision to publish, or preparation of the manuscript."

Please include your amended Funding Statement and Role of Funder statement within your cover letter. We will change the online submission form on your behalf.

Reviewers' comments:

Reviewer's Responses to Questions

**Comments to the Author**

1. Is the manuscript technically sound, and do the data support the conclusions?

Reviewer #1: Yes

Reviewer #2: Yes

2. Has the statistical analysis been performed appropriately and rigorously? 

Reviewer #1: Yes

Reviewer #2: No

3. Have the authors made all data underlying the findings in their manuscript fully available?

Reviewer #1: Yes

Reviewer #2: No

4. Is the manuscript presented in an intelligible fashion and written in standard English?

Reviewer #1: Yes

Reviewer #2: No

5. Review Comments to the Author

Reviewer #1: The study provides a comprehensive dataset of kinematic data of the body in an eating task for a cohort of healthy individuals. The dataset is useful for biomechanical and clinical research (although I don’t think this has been articulated particularly well in the discussion section). The methods used are appropriate but sometimes lack clarity.

My main comments are related to the discussion section, I don’t feel that it has discussed the important aspects of having a dataset like this to a great extent and instead focussed on the differences between the eating phases. Please see my comments in detail below.

I am not clear on what what S3 and S4 are showing. Would the authors also be willing to make the raw data available to the readers? I understand that this is difficult to upload as an excel sheet because of the volume, but perhaps a statement to get in touch with the authors or placing them in an open repository would be useful.

Abstract line 30 -34

The following sentence is quite vague “The averaged shoulder, elbow, and hip joint flexion angles were significantly the largest in the mouth phase, with the smallest being the neck flexion angle “, It is also not clear why you would compare the various joint rotations to each other i.e. what is the clinical or otherwise importance of this?

The results and conclusion sections of the abstract needs rewording, the main finding of ththe study are not clear from the sentences here.

The introduction section is written clearly and is well structured.

The methods

Line 78 – not clear why you have this age range (looks like a very narrow range), why only include right-handed individuals ?

Line 122- 127 – The paper is based on the premise that it would analyse natural movement during eating, yet there are a series of restrictions on how the movement is conducted. I understand that this is a necessity in studies of this sort, but for example, it is not clear to me what is an excessive upper limb elevation and why that would be excluded? It begs the question – are you artificially reducing the variation in movement?

Can you also clarify whether participants were made aware of these 4 phase and restrictions? Or were these simply filtered afterwards? Can you provide a clear description of the instructions given to the participants?

Lines 133-134

It is not clear why and what is meant by “Phases in which the difference between the minimum and maximum joint angle was >5.0° were identified”

Results

Line 157 – why were 5 excluded?

Table 3 provides a summary of the differences between phases, I am struggling to see how this information can be useful. It is expected that there would be differences between the phases. I would have much preferred plots of the rotations of the 4 phases and a visual representation of the change in angles i.e Figure 1 .

Discussion

The discussion section is focussed on the differences between the phases, again I am struggling here to see how this would be useful clinically or otherwise (biomechanical modelling for example). I would have expected more in-depth discussion of the following:

- Differences between individuals and whether there were clear differences in movement strategy leading to the large variation shown in Fig 1

- How the movements compare to children and older adults * you made the reference to these group earlier

- How the movements compare to pathological populations

- The limitations of these methods and data in identifying deviations from normal movement because of the large variability

- Limitations related to applying this experimental protocol with strict instructions to individuals with impairments.

Reviewer #2: The purpose of this study was to establish normative movement kinematics of the upper extremity, trunk and hip during realistic eating with a spoon. The major issue with the paper is the lack of sufficient motivation or gap they are filling. The authors argue that little work has been done to quantify motion when people actually eat versus simulate eating (though other studies have done this) and that no studies explore lower limb kinematics. As the individuals do not use their lower limbs when sitting to eat, it is hard to understand why this gap is important. The introduction could do more to establish the gaps of prior findings and to remove portions that repeat (like the end of paragraph 1). The final paragraph also indicates that an area of focus should be on the waveform analysis or analysis over time. It is unclear what either statement means as no waveforms are included (no figures or assessment of the trajectory) and this is done at a single point in time. If this refers instead to splitting analysis into the phases of movement, that should be more clear.

The lack of clear question also contributes to a lack of focus in the outcome measures and description. It is not clear why the authors measure the mean for a phase in addition to the range of motion. It is also unclear why the phases are statistically compared to one another (i.e. what does it mean that the mean is significantly higher for spooning versus reaching?). Presenting range of motion seems that it would be sufficient to state which motion is used more.

Specific Comments

Abstract, Line 33-34: The sentence “Our results revealed that coordination of whole-body movement patterns correspond to each realistic eating phase in healthy individuals” does not make sense as the authors did not quantify coordination

Introduction, Line 46: ‘rehabilitation’ not ‘rehabilitating’

Line 54:suggest replacing ‘normal’ to ‘for an unimpaired population’

Line 112: It is unclear how the task was separated into phases (i.e. was this done purely visually or using a threshold for some variable?)

The description of the imu system set-up should be more detailed and provide accuracy measures (line 96) for each angle measured. In particular, it is unclear what the angle is in the spine measured between only two segments (e.g. C7T1) or if this with respect to global? Assuming the latter, but it is unclear.

Line 139: it is unclear what ‘integrated for each phase’ means as no areas are presented.

Line 152: Effect size descriptions should use ranges rather than equal signs ‘>=’ rather than ‘=’

The hip angles are quite small (50 deg). Was the chair height standardized to each participant? Could they keep their feet fully on the floor?

Line 157: suggest ‘participants’ rather than ‘patients’

Line 160: What is meant by a rest period?

Suggest ‘mean angle’ rather than AVA

Results would benefit from a figure of the trajectory rather than simply tables.

Suggest removing all stats comparing phases as the purpose is not clear

The discussion notes that the kinematic model is not validated, while the methods indicate that it is. Which is correct?

6. PLOS authors have the option to publish the peer review history of their article (what does this mean?). If published, this will include your full peer review and any attached files.

Reviewer #1: No

Reviewer #2: No

---

## [Author Response · Author response to Decision Letter 0]

19 Jul 2021

Manuscript ID: PONE-D-20-39538

Title: Exploring whole-body kinematics when eating real foods with the dominant hand in healthy adults

Point-by-Point Response

Thank you for reviewing our manuscript and providing thoughtful comments. We have revised our manuscript accordingly. Please note that the changes made do not influence the content, conclusions, or framework of the paper. We have not listed below all minor changes made; however, these can be viewed as “tracked changes” in the revised manuscript.

Academic Editor: Critique and Responses

Comment: 1. Please ensure that your manuscript meets PLOS ONE's style requirements, including those for file naming.

Response: The supporting file title names were changed on page 29, and the file names are now “S1_Table–S7_Table.”

Comment: 2. In your Methods section, please provide additional information about the participant recruitment method and the demographic details of your participants. Please ensure you have provided sufficient details to replicate the analyses such as: aa) a description of how participants were recruited, and b) descriptions of where participants were recruited and where the research took place.

Response: The following text was added: 

page 5 line 86: “…among the medical staff of our institution, who responded of their own free will to a request for study participation” 

page 6 line 94 “…in the rehabilitation room of our institution.”

Comment: 3. Please provide a sample size and power calculation in the Methods, or discuss the reasons for not performing one before study initiation.

Response: In the Discussion (page 21, line 347), the following sentence was added: “The current study had no preplanned sample size and the study power had not been calculated because the significance level and power of differences in measured joint angle data between phases were not known prior to the study.

Comment: 4. We note that Figure 1 includes an image of a participant in the study. ~If you are unable to obtain consent from the subject of the photograph, you will need to remove the figure and any other textual identifying information or case descriptions for this individual.

Response: We have removed the images of the participant from the manuscript, as we could not obtain written consent from the subject of the photograph.

Comment: 5. Thank you for stating in your Funding Statement: "JN was partly funded by a grant from University of Miyazaki hospital. The funder had a role in preparation of the manuscript."

5.1. Please provide an amended statement that declares *all* the funding or sources of support (whether external or internal to your organization) received during this study, as detailed online in our guide for authors at http://journals.plos.org/plosone/s/submit-now. Please also include the statement “There was no additional external funding received for this study.” in your updated Funding Statement.

5.2. Please state what role the funders took in the study. If the funders had no role, please state: "The funders had no role in study design, data collection and analysis, decision to publish, or preparation of the manuscript."

Please include your amended Funding Statement and Role of Funder statement within your cover letter. We will change the online submission form on your behalf.

Response: We have provided an amended Funding Statement and Role of Funder statement within the cover letter.

Reviewer #1: Critique and Responses

Comment: I am not clear on what what S3 and S4 are showing. 

Response: Tables S3 and S4 (revised S5) show the mean angles and ROMs, respectively, of each eating phase for each participant. The table titles were added on page 29.

Comment: Would the authors also be willing to make the raw data available to the readers? I understand that this is difficult to upload as an excel sheet because of the volume, but perhaps a statement to get in touch with the authors or placing them in an open repository would be useful.

Response: We agree that the availability of the raw kinematic data in public repositories would be valuable as you suggested. We would like to consider this point positively in the future.

Comment: Abstract line 30 -34

The following sentence is quite vague “The averaged shoulder, elbow, and hip joint flexion angles were significantly the largest in the mouth phase, with the smallest being the neck flexion angle “, It is also not clear why you would compare the various joint rotations to each other i.e. what is the clinical or otherwise importance of this? The results and conclusion sections of the abstract needs rewording, the main finding of the study are not clear from the sentences here.

Response: The aim of the study was clarified (page 2, line 20) as follows: “…by assessing movement patterns in defined phases.” 

The Methods and Results were revised as follows: 

“The mean joint angles were compared among the phases with Friedman’s analysis of variance.” (page 2, line 25)

“The mean shoulder, elbow, and hip joint flexion angles were largest in the mouth phase, with the smallest being the neck flexion angle. By contrast, in the spooning phase, the shoulder, elbow, and hip flexion were the smallest, with the largest being the neck flexion angle. These angles were significantly different between the mouth and spooning phases (p < 0.008, Bonferroni post hoc correction).” (page 2, line 27)

The following sentences were modified and added to the conclusion: “Our results revealed that characteristic whole-body movements correspond to each phase of realistic eating in healthy individuals. This study provides useful kinematic data regarding normal eating movements, which may inform whole-body positioning and movement, improve the assessment of eating abilities in clinical settings, and provide a basis for future studies. (page 2, line 32)

Comment: The methods

Line 78 – not clear why you have this age range (looks like a very narrow range), why only include right-handed individuals ?

Response: The following words were added to the participants section: “To ensure normal kinematic data and exclude age effects resulting from neurological immaturity or decline, children [6,16] and older adults [17] were not selected as their upper limb reaching movements during activities of daily living differ from those of young adults. Left-handers were excluded since manipulations of their dominant hand may not simply mirror those of right-handers [18].” (page 5, line 87)

Comment: Line 122- 127 – The paper is based on the premise that it would analyse natural movement during eating, yet there are a series of restrictions on how the movement is conducted. I understand that this is a necessity in studies of this sort, but for example, it is not clear to me what is an excessive upper limb elevation and why that would be excluded? It begs the question – are you artificially reducing the variation in movement?

Response: The following words were added to the data processing section: 

“This cycle was chosen as a realistic eating movement according to the study aims, but was not completely realistic because using the non-dominant hand to hold the bowl or to gather food was not permitted.” (page 8, line 131)

“Cycles containing the following motions were not accepted because they conflict with our definition of an eating movement: excessive elevation of the upper limb during transport in an exaggerated or unnecessary manner, looking away from the bowl, …”(page 8, line 136)

Comment: Can you also clarify whether participants were made aware of these 4 phase and restrictions? Or were these simply filtered afterwards? Can you provide a clear description of the instructions given to the participants?

Response: These following sentences were modified or added: ‘Participants received the task instruction “please eat without a break three spoonfuls of yogurt with the right hand, using habitual whole-body movements and speed, not looking away, with the left hand resting on the left thigh.” (page 7, line 121)

“These phases were subsequently filtered to identify phase recordings that were suitable for further analyses.” (page 8, line 130)

Comment: Lines 133-134

It is not clear why and what is meant by “Phases in which the difference between the minimum and maximum joint angle was >5.0° were identified”

Response: The sentence was revised as follows: “Joint motion directions, in which the difference between the minimum and maximum joint angles was >5.0°, were identified for further analyses as these are clinically detectable.” (page 9, line 149)

Comment: Results

Line 157 – why were 5 excluded?

Response: The following sentence was revised in the data processing section: “Cycles containing the following motions were not accepted because they conflict with our definition of an eating movement: …” (page 8, line 136)

Comments: Table 3 provides a summary of the differences between phases, I am struggling to see how this information can be useful. It is expected that there would be differences between the phases. I would have much preferred plots of the rotations of the 4 phases and a visual representation of the change in angles i.e Figure 1 .

Response: The tables were changed to Figs 1 and 2 with Table S4 and S6 provided as supporting files.

Comment: Discussion

The discussion section is focussed on the differences between the phases, again I am struggling here to see how this would be useful clinically or otherwise (biomechanical modelling for example). I would have expected more in-depth discussion of the following:

- Differences between individuals and whether there were clear differences in movement strategy leading to the large variation shown in Fig 1

Response: The following paragraph was added to the Discussion: “Inter-subject differences may have contributed to the high variability in joint angles shown in Fig 3, such that the SD values of all joint angles and phases do not converge. There was no clear difference in movement strategies among the four defined eating phases analyzed in this study, even after the exclusion of outliers with excessive deviation. However, as the aim of the study was to analyze habitual whole-body movements, movements at the participant’s own speed without constraints from a backrest were allowed, and the seat height, as well as the bowl and spoon sizes, were not adapted to the participant’s body size. The start and stop positions were also defined within uninterrupted eating movements. Thus, measurements of realistic eating behavior can easily vary. The influence of body constitution and muscle strength, which may have caused movement differences in the current study, should be examined in clinical settings by future studies. In contrast, a comparison of eating phases could foster an understanding of typical whole-body movement patterns in a realistic condition such as that described above, allowing movement patterns that reflect realistic eating to be adopted in clinical practice.” (page 19, line 310)

Comment: - How the movements compare to children and older adults * you made the reference to these group earlier

Response: The following sentences were added to the Discussion: “Increased maximal shoulder abduction and reduced maximal elbow flexion of healthy pediatric participants compared to those of healthy young adults were reported previously [6]. Since these maximal angles were shown during the mouth phase in our study with healthy young adults, larger shoulder abduction and lower elbow flexion may be able to be assessed as normal pediatric motions in the mouth phase. Although the effects of aging on eating movements remain unclear, the reaching movement during eating may be affected in older individuals. Healthy older adults have an increased number of jerky and undirected movements during reaching tasks [17], possibly resulting in increased joint angles of the upper limb compared to our results.” (page 20, line 324)

Comment: - How the movements compare to pathological populations

Response: The following sentences were revised in the Discussion: “Other studies have reported greater movement of the trunk and elbow flexion during a drinking task among individuals who have had a stroke compared to controls; however, reports of between-group comparisons during eating have been lacking [30,31]. The trunk position is supported by the lower limbs in seated eating, and the control of the lower limbs is accomplished in the normal hip joint angles. The regulation of the hip angles and the reduction in elbow extension during eating should therefore be investigated in future studies. Although the hip angle varied little among our study group, the compensatory coordination of this position along with reduced elbow extension could have an important clinical effect on eating in populations with neurological problems.” (page 16, line 262)

Comment: - The limitations of these methods and data in identifying deviations from normal movement because of the large variability

Response: The following sentences were revised in the limitations section: “We constrained the task by not allowing the individual to hold the bowl, only used yogurt, and excluded some movements like the excessive raising of upper limb during the transporting phase that was not predefined in the study task. Although this standardized our assessment, realistic eating movements still had wide variability as described above. This limitation is important to be considered for any clinical application of our study results. The findings also may not be generalizable to other types of food, utensils other than a spoon, and other eating movements, such as stirring, cutting, and gathering food. (page 20, line 333)

Comment: - Limitations related to applying this experimental protocol with strict instructions to individuals with impairments.

Response: The following sentence was added to the limitations section: “Clinical assessments according to our experimental protocol should be considered, with guidance to use the strict instructions demonstrated in the study.” (page 20, line 341) 

Reviewer #2: Critique and Responses

Comment: The major issue with the paper is the lack of sufficient motivation or gap they are filling. The authors argue that little work has been done to quantify motion when people actually eat versus simulate eating (though other studies have done this) and that no studies explore lower limb kinematics. As the individuals do not use their lower limbs when sitting to eat, it is hard to understand why this gap is important. The introduction could do more to establish the gaps of prior findings~

Response: The following words were added to the Introduction: “whose position as a stable base of upper limb motion should be assessed in practice” (page 4, line 59)

Comment: and to remove portions that repeat (like the end of paragraph 1).

Response: We removed the following text from the Introduction: “Of these studies, only Beaudette and Chester [6] performed the eating task under realistic conditions.” 

Comment: The final paragraph also indicates that an area of focus should be on the waveform analysis or analysis over time. It is unclear what either statement means as no waveforms are included (no figures or assessment of the trajectory) and this is done at a single point in time. If this refers instead to splitting analysis into the phases of movement, that should be more clear.

Response: The following underlined text was added to the last paragraph of the Introduction: 

“Waveform analyses of upper limb joint movements, which provide only a subjective visual assessment of upper limb motion for eating, have been previously reported [6,11,13].... Moreover, to the best of our knowledge, no comparison of single points or phases during eating in regards to objective parameters of whole-body kinematics have been reported. Realistic eating consists of specific phases that require concurrent whole-body movements. Due to the complexity of these realistic movements, clinicians in practice may have a limited understanding of the normal kinematics of eating. (page 4, line 62)

Comment: The lack of clear question also contributes to a lack of focus in the outcome measures and description. It is not clear why the authors measure the mean for a phase in addition to the range of motion. It is also unclear why the phases are statistically compared to one another (i.e. what does it mean that the mean is significantly higher for spooning versus reaching?). Presenting range of motion seems that it would be sufficient to state which motion is used more.

Response: The following underlined text was added to the last paragraph of the Introduction: “Establishing whole-body kinematic variables and movement patterns, which could be described as differences in joint positions, the extent of motion, and motion directions between eating phases in patients with eating limitations would therefore constitute useful information that could be applied in clinical settings. Thus, the aim of our study was to determine the whole-body joint angles and movement patterns necessary for realistic eating in healthy individuals.” (page 4, line 69)

Similarly, the following sentences were modified in the data processing section : “ To compare and characterize the joint angles of each phase, the following variables were calculated: mean angles, calculated from the cumulative sequential data over each phase to provide the average position; range of motion (ROM), calculated as the difference between the minimum and maximum angles in each phase to quantify the motion [27]; and joint angles normalized to 101 points, corresponding to 0–100% of the movement time in each phase. The normalized data were then represented as the mean (SD) for each point in each phase, and waveforms of normalized joint angles were calculated for the visual analysis of the joint motion direction. (page 9, line 151)

Comment: 

Abstract, Line 33-34: The sentence “Our results revealed that coordination of whole-body movement patterns correspond to each realistic eating phase in healthy individuals” does not make sense as the authors did not quantify coordination

Response: The following underlined text was modified: “Our results revealed that characteristic whole-body movements correspond to each phase of realistic eating in healthy individuals.” (page 2, line 32)

Comment: Introduction, Line 46: ‘rehabilitation’ not ‘rehabilitating’

Response: The word “rehabilitating” was changed to “rehabilitation” on page 3, line 45.

Comment: Line 54:suggest replacing ‘normal’ to ‘for an unimpaired population’

Response: The sentence was revised as follows: “The upper limb joint angles in an unimpaired population when eating with a spoon have previously been reported using three-dimensional motion analysis [6–13].” (page 4, line 54)

Comment: Line 112: It is unclear how the task was separated into phases (i.e. was this done purely visually or using a threshold for some variable?)

Response: The following words were added to the data processing section: “…by visually partitioning the recorded movie:…” (page 8, line 127)

Comment: The description of the imu system set-up should be more detailed and provide accuracy measures (line 96) for each angle measured. In particular, it is unclear what the angle is in the spine measured between only two segments (e.g. C7T1) or if this with respect to global? Assuming the latter, but it is unclear.

Response: The following sentence was added to the instruments and measurements section: “Of these points, one neck, four spine, and two toe points were not directly measured but were calculated using ISU data and other data points with respect to global and local references.” (page 6, line 101)

Comment: Line 139: it is unclear what ‘integrated for each phase’ means as no areas are presented.

Response: The following text was modified in the data processing section: “The normalized data were then represented as the mean (SD) for each point in each phase, and waveforms of normalized joint angles were calculated for the visual analysis of the joint motion direction.” (page 9, line 156)

Comment: Line 152: Effect size descriptions should use ranges rather than equal signs ‘>=’ rather than ‘=’

Response: Accordingly, the description was revised from |r| = 0.1 to |r| = 0.10–0.29, from |r| = 0.3 to |r| = 0.30–0.49, and from |r| =0.5 to |r| ≥ 0.50 (page 11, line 169).

Comment: The hip angles are quite small (50 deg). Was the chair height standardized to each participant? Could they keep their feet fully on the floor?

Response: The following underlined words were added to the instruments and measurements section: “After calibration and definition of body dimensions, participants sat comfortably without trunk fixation to allow free motion on a 40-cm high seat behind a table (10 cm from the participant’s trunk, at the height of the right elbow), with the right upper limb placed alongside the body at baseline and their feet fully on the floor.” (page 7, line 113)

Comment: Line 157: suggest ‘participants’ rather than ‘patients’

Response: The word “patients” was revised to “participants” (page 11, line 175).

Comment: Line 160: What is meant by a rest period?

Response: The following words were added to the study sample section: “…, i.e., the reaching movement paused within several seconds and started again,…” (page 11, line 178)

Comment: Suggest ‘mean angle’ rather than AVA

Response: All terms containing “AVA” were revised to “mean angle.”

Comment: Results would benefit from a figure of the trajectory rather than simply tables.

Response: The tables presenting the mean angles and ROMs of the eating phases were changed to figures Fig 1 and Fig 2.

Comment: Suggest removing all stats comparing phases as the purpose is not clear

Response: All p-values were removed, and only the significance levels (e.g., p < 0.008) are provided in the Results and the legends of Figs 1 and 2.

Comment: The discussion notes that the kinematic model is not validated, while the methods indicate that it is. Which is correct?

Response: To enhance clarity for the reader, the last sentence of the Discussion was removed.

---

## [Decision Letter · Decision Letter 1]

27 Sep 2021

PONE-D-20-39538R1Exploring whole-body kinematics when eating real foods with the dominant hand in healthy adultsPLOS ONE

Dear Dr. Nakatake,

Thank you for submitting your manuscript to PLOS ONE. After careful consideration, we feel that it has merit but does not fully meet PLOS ONE’s publication criteria as it currently stands. Therefore, we invite you to submit a revised version of the manuscript that addresses the points raised during the review process.

We look forward to receiving your revised manuscript.

Kind regards,

Bernadette Ann Murphy, PhD

Academic Editor

PLOS ONE

Journal Requirements:

Additional Editor Comments (if provided):

There are a couple of minor corrections needed to finalize the manuscript.

Reviewers' comments:

Reviewer's Responses to Questions

**Comments to the Author**

1. If the authors have adequately addressed your comments raised in a previous round of review and you feel that this manuscript is now acceptable for publication, you may indicate that here to bypass the “Comments to the Author” section, enter your conflict of interest statement in the “Confidential to Editor” section, and submit your "Accept" recommendation.

Reviewer #1: (No Response)

2. Is the manuscript technically sound, and do the data support the conclusions?

Reviewer #1: Yes

3. Has the statistical analysis been performed appropriately and rigorously? 

Reviewer #1: Yes

4. Have the authors made all data underlying the findings in their manuscript fully available?

Reviewer #1: Yes

5. Is the manuscript presented in an intelligible fashion and written in standard English?

Reviewer #1: Yes

6. Review Comments to the Author

Reviewer #1: The authors have made significant improvement to the manuscript and have attamepted to address all the comments made by the reviewers.

There is still limited justification for the the compasiron between the four different phases - it does not seem necessary and it is not clear what added value is gained from doing this. I think the presentation of movement patterns and ranges of motion in the different phases is what is important here and I was glad to see the new figures that replaced the tables. Nevertheless, the comparison between the different phases have been considerably watered down in this version and are not made to be the focus which is good in my opinion.

One minor comment: The sentence describing the instructions given to the participants is quite complex, was this the language used to describe this? Or does it sound complex because it is a literal translation of the instructions?

7. PLOS authors have the option to publish the peer review history of their article (what does this mean?). If published, this will include your full peer review and any attached files.

Reviewer #1: No

---

## [Author Response · Author response to Decision Letter 1]

13 Oct 2021

Manuscript ID: PONE-D-20-39538R1

Title: Exploring whole-body kinematics when eating real foods with the dominant hand in healthy adults

Point-by-Point Response

Thank you for reviewing our manuscript and providing thoughtful comments. We have revised our manuscript accordingly. Please note that the changes made do not influence the content, conclusions, or framework of the paper. We have not listed below all minor changes made; however, these can be viewed as “tracked changes” in the revised manuscript.

Reviewer #1: Critique and Responses

Comment: There is still limited justification for the comparison between the four different phases - it does not seem necessary and it is not clear what added value is gained from doing this. I think the presentation of movement patterns and ranges of motion in the different phases is what is important here and I was glad to see the new figures that replaced the tables. Nevertheless, the comparison between the different phases have been considerably watered down in this version and are not made to be the focus which is good in my opinion.

Response: The following underlined parts were added to the Discussion section.

“The different movements from this pattern, especially the reaching phase, and the mean angles of the shoulder internal rotation and the wrist flexion were the smallest. The spooning phase also required large ROMs of forearm pronation (decreasing angles) as well as of wrist flexion (increasing and decreasing angles) and radial deviation (decreasing angles). Knowledge of the movement patterns that we have described may be useful for the comprehensive assessment of eating and optimization of upper and lower limb, neck, and trunk movements during eating. To facilitate or compensate for the normal positions and movements for each eating phase, appropriate positioning, seating equipment, upper extremity and posture control, and orthotics [3, 4] referring to the mean angles, the ROMs, and the changes in joint angles of the present study could be provided in practice.” (page 18)

“Maximal shoulder flexion and abduction angles in individuals with rotator cuff impingement are approximately the same as those in healthy individuals [7]; thus, these two groups’ shoulder movements would not be different in the mouth phase examined here. The severe upper extremity function in children with unilateral cerebral palsy and their maximal forearm pronation angle during eating are positively correlated [9], which could be observed in our spooning phase.” (page 20)

Comment: The sentence describing the instructions given to the participants is quite complex, was this the language used to describe this? Or does it sound complex because it is a literal translation of the instructions?

Response: The following were the approximate instructions provided to the participants. Underlined sections underwent minor revisions.

“Participants held a spoon with their right hand and received the task instruction, “please eat three spoons full of yogurt without a break, using habitual whole-body movements and speed, not looking away, and with your left hand resting on your left thigh.” (page 7)

---

## [Editor Report · Decision Letter 2]

15 Oct 2021

Exploring whole-body kinematics when eating real foods with the dominant hand in healthy adults

PONE-D-20-39538R2

Dear Dr. Nakatake,

We’re pleased to inform you that your manuscript has been judged scientifically suitable for publication and will be formally accepted for publication once it meets all outstanding technical requirements.

Kind regards,

Bernadette Ann Murphy, PhD

Academic Editor

PLOS ONE
---

## [Editor Report · Acceptance letter]

20 Oct 2021

PONE-D-20-39538R2 

Exploring whole-body kinematics when eating real foods with the dominant hand in healthy adults 

Dear Dr. Nakatake:

I'm pleased to inform you that your manuscript has been deemed suitable for publication in PLOS ONE. Congratulations! Your manuscript is now with our production department. 

Kind regards, 

on behalf of

Dr. Bernadette Ann Murphy 

Academic Editor

PLOS ONE